# Cortisol response under low intensity exercise during cognitive-behavioral therapy is associated with therapeutic outcome in panic disorder–an exploratory study

**Gloria-Beatrice Wintermann**[ORCID]*, **René Noack, Susann Steudte-Schmiedgen, Kerstin Weidner**

Department of Psychotherapy and Psychosomatic Medicine, University Hospital Carl Gustav Carus Dresden, Technische Universität Dresden, Dresden, Saxony, Germany

* gloria.wintermann@uniklinikum-dresden.de

## Abstract

### Objectives

Patients with Panic Disorder (PD) show an abnormal stress-induced functioning of the Hypothalamic-Pituitary-adrenal (HPA)-axis. Different protocols for stress induction are of rather low relevance for the psychotherapeutic treatment. In practice, interoceptive exposure is often realized as Low Intensity Exercise (LIE), as compared to an incremental cycle exercise test to exhaustion. Currently, it is not known, whether LIE displays an effective interoceptive stressor 1.) leading to a significant anxiety induction; 2.) a comparable HPA- and Sympathetic-Adreno-Medullar (SAM)-axis response in both patients and healthy controls; 3.) stress responses under LIE are associated with treatment outcomes.

### Patients and methods

N = 20 patients with PD and n = 20 healthy controls were exposed to ten minutes of LIE on an exercise bike. LIE was applied as part of the interoceptive exposure, during an intensive Cognitive-Behavioral Therapy (CBT) in a day clinic. Heart rate was monitored and salivary cortisol samples collected. Before and after the LIE, state anxiety/ arousal were assessed. In order to evaluate psychopathology, the Panic and Agoraphobia Scale, Mobility Inventory, Agoraphobic Cognitions Questionnaire and Body Sensations Questionnaire were applied, before (T1) and after five weeks (T2) of an intensive CBT.

### Results

LIE led to a significant and similar heart rate increase in both groups. Cortisol decreased over time in both groups, especially in male patients. A higher psychopathology before, and after CBT, was associated with a significantly lower cortisol response under LIE.

**Data Availability Statement:** The data file and study protocol are available from the Open Science

Framework (https://archive.org/details/osf-registrations-s76rw-v1, source https://osf.io/s76rw/, registration DOI 10.17605/OSF.IO/S76RW and identifier DOI 10.17605/OSF.IO/NE4G6).

**Funding:** The present study was funded by the Robert-Pfleger-Stiftung. GBW received the award. The funders had no role in study design, data collection and analysis, decision to publish, or preparation of the manuscript.

**Competing interests:** The authors have declared that no competing interests exist.

## Conclusions

In the present study, LIE led to a divergent stress response: while there was a significant heart rate increase, cortisol decreased over time, particularly in male patients. A lower reactivity of the HPA-axis seems to be associated with a lower treatment outcome, which may affect extinction based learning. The findings suggest, that interoceptive stimuli should be designed carefully in order to be potent stressors.

## Introduction

Panic disorder (PD) with or without agoraphobia is a common psychological disorder with a life-time prevalence of 4.8% [1]. Patients experience physical and psychological signs like sudden palpitations, dizziness, sweating, dyspnoe, anxious worrying and avoidance, among others. These symptoms resemble physical sensations under stress exposure [2]. It has been shown that stress plays a pivotal role in the etiopathogenesis of PD. For instance, studies give evidence for the first onset of panic attacks after stressful life-events [3, 4]. An abnormal functioning of the stress axes under different stressors is supposed, which may increase the risk for the evolvement and persistence of PD symptoms [5–7]. In line, former studies investigated abnormalities of the main stress axes, including both the Sympathetic-Adreno-Medullar (SAM) system and the Hypothalamic-Pituitary-adrenal (HPA)-axis, under different stimulations. Findings could show differential secretion patterns of the main secretagogues and secretory end products such as the Corticotropin-Releasing-Hormone (CRH), Adrenocorticotropic Hormone (ACTH), alpha-amylase or cortisol [for a systematic review: 8]. Most studies point towards an increased cortisol level before stress exposure [6, 9, 10], dissociative response patterns between the stress axes [11], and an insufficient cortisol response under different stressors, like an oral presentation [12] or hormonal stress induction [13, 14].

However, the results vary widely depending on methodological differences and peculiarities of the specific patient sample, among them, the illness duration, age of onset, extent of former psychopharmacological/ psychotherapeutic treatment, personality traits and, not least, the kind or intensity of the stress exposure [systematic reviews: 8, 15]. Above, present studies used psychosocial stress protocols such as the Trier Social Stress Test (TSST) [11, 12], hormonal stimulation tests such as the Dexamethason-CRH test [16] or applied panicogenic agents [for a review: 17]. These stress protocols are cost-intensive and of rather low relevance for the treatment of patients with PD. According to the current psychotherapy guidelines for anxiety disorders, repeated exposure to feared external and internal/ interoceptive cues has been shown to lead to the highest effect sizes in the treatment response in these patients [18]. In particular, interoceptive exposure is associated with better treatment efficacy and acceptability [19]. Typically, a catastrophic misinterpretation of bodily sensations takes place in PD [20]. Above, patients present an increased sensitivity towards physical sensations of anxiety and fear [21]. Following, at the beginning of the exposure therapy, patients are repeatedly exposed to physical/ interoceptive stimuli. For instance, hyperventilation or bicycle ergometry tasks are applied [22]. However, in practice, the interoceptive exposure lacks standardization and activates dysfunctional, defensive mobilization, especially in highly anxiety sensitive patients [23]. Following, interoceptive exposure is often realized as Low Intensity Exercise (LIE), as compared to an incremental bicycle ergometry test to exhaustion or a supramaximal bicycle ergometry test [24]. Although, LIE seems to be a suitable and economically valid stimulus of low costs, it displays a less intensive form of interoceptive exposure (when compared with bicycle ergometry)

and may be associated with lower reductions in anxiety sensitivity as well as fearful responding [25].

Present findings hint toward the necessity of a sufficient stress response in order to benefit from exposure therapy in the long-run [6, 26]. It is well-known that only exercise of high intensity and long duration results in a significant activation of the stress axes in healthy subjects. In line, bicycle ergometry has been shown to reliably activate both the SAM- and HPA-axis, when at least the aerobic/anaerobic threshold is reached [27–30]. In contrast, low intensity exercise (40% VO$_2$max) does not result in significant increases in circulating cortisol levels, but even in a decrease [29]. Currently, there is a lack of findings regarding the suitability of LIE as interoceptive exposure. Furthermore, it is not clear whether the stress response under LIE is associated with a significant activation of the stress axes and with the treatment response in PD.

Therefore, the first aim of the present study was to investigate whether patients with PD with/ without agoraphobia show a significant anxiety induction under LIE; second, whether they show a significant HPA-/ SAM-axis response as compared with healthy subjects. Third, it was of interest, whether the stress reactivity under LIE is associated with the treatment outcome in PD. The results could be of clinical relevance, since a lack of an appropriate stress response might necessitate the application of other kinds of symptom provocations, a higher standardization of interoceptive exposure protocols and a more careful planning, in order to appropriately trigger panic-like symptoms.

## Material and methods

### Ethics statement

The present study is an exploratory one which was performed as part of another study [31], investigating the effect of perceiving one´s own stress-related body odors on the neural activity in patients with PD (for more details see the study protocol in the supplementary material). The study was approved by the University of Dresden Medical Faculty Ethics Review Board (EK: 24022009) and fits the recommendations of the Declaration of Helsinki on Biomedical Research Involving Human Subjects. After description and explanation of the complete study protocol (see S1 and S2 Files), participants signed in a written informed consent.

### Study registration

The present trial has been registered in the German Clinical Trials Register (DRKS00028117), after enrollment of the participants started. The reason for the retroactive registration is that the present trial was part of a larger study without clear clinical reference. In the latter study, the effect of perceiving one´s own stress-related body odors on the neural activations in a fear-network was investigated. For this purpose, sweat odors were collected during a standardized, psychosocial stress test and during a LIE). The participants wore odorless T-shirts during both stress conditions (for a detailed description of the study protocol: see study protocol in the S1, S2 Files and [31]). The study protocol has been published at protocols.io [http://dx.doi.org/10.17504/protocols.io. eq2lynbervx9/v1]. The performance during the LIE was correlated with questionnaire data, gained during routine quality control of a cognitive-behavioral therapy (CBT) in a day-clinic specialized on the treatment of PD with/ without agoraphobia. The authors confirm that all ongoing and related trials for this intervention are registered in the German Clinical Trials Register

### Study participants

The patients were continuously recruited as convenience sample in an outpatient unit of a clinic for psychosomatic medicine and psychotherapy at a large University Hospital, from

2009 on. The outpatient unit was specialized on the assessment and treatment of patients with panic symptoms and closely cooperated with a day clinic which patients were referred to in case of severe impairment.

Patients were assessed according to a psychological morbidity, using the Structured Clinical Interview (SCID) [32] for the Diagnostic and Statistical Manual of Mental Disorders (DSM-IV) axis I and II disorders [2]. Patients with a primary diagnosis of PD with/ without agoraphobia, fluency in the German language, recommended for an intensive CBT in a day clinic, willingness to participate in the present study (including several parts as has been described elsewhere [31], among them a LIE with parallel heart rate monitoring as well as saliva cortisol sampling) were included in the present study.

Patients with a secondary diagnosis of dysthymia or mild depression as well as patients with habitual cigarette smoking ($\leq$ 10 cigarettes/ day) and oral contraceptives in females were not precluded from participation in order to avoid selection bias. Exclusion criteria were: any other life-time psychological disorders, psychopharmacological drug treatment and any severe medical illness (e.g. cancer, metabolic/ autoimmune/ endocrinological/ cardiovascular disorders). The healthy individuals were recruited by public advertisements and matched by age and sex to the patient sample. They were not allowed to have a history of a mental disorder. This was confirmed using the SCID-I interview [32]. After the ascertainment of the diagnosis of PD with/ without agoraphobia and rating the functional impairment to be at least moderate, patients were offered a CBT in a psychosomatic day-clinic [33].

Participants received a compensation of 150 Euro when they took part in the whole study procedure including a psychosocial stress test, physiological stimulation test, LIE and functional magnetic resonance imaging (fMRI) [31].

The sample size was determined using G*Power 3.1.9.2 [34]. For the calculation of analyses of variance for repeated measures and within-between interactions for two groups (patients vs. healthy controls), a medium effect size of f = .25, a type I error probability of 5% and a statistical power (1-β) of 80% were assumed. The mentioned assumptions led to a total a priori sample size of 20 participants.

## Low intensity exercise

All study participants took part in a LIE during their stay at a day clinic specialized on the treatment of panic disorder/ agoraphobia. The LIE was realized as interoceptive exposure, once during an intensive 5-week CBT. The decisions to use a LIE intervention were made in the context of routine clinical care, mainly by the former director (Katja Petrowski) of our psychosomatic day clinic. The latter was specialized on the treatment of anxiety disorders. It took place on average at the third week, after the beginning of the CBT. The reason why the exercise test was done several weeks into therapy was that female participants were tested during the luteal phase of their menstrual cycle. The LIE included cycling on an exercise bike, with a duration of ten minutes, a constant resistance of ten Watt, a minimum of 110 bpm and a maximum of 120 bpm [35, 36] and guided by a certified psychologist (GBW). In order to boost anxiety, all patients were asked to direct their attention towards anxiety-related body cues (heart rate, breath). In order to reduce circadian variations in cortisol concentrations, the testing procedures took place between three pm and six pm [11]. Participants were asked to refrain from smoking, alcohol consumption, eating and intense physical exercise at least 2 hours prior to the start of the testing.

## Cognitive-behavioral therapy (CBT)

All patients were treated within a standardized, manual-based therapy according to Lang, Helbig-Lang [33] and as has been described by Wichmann et al. [26]. During the five weeks of

intensive CBT at a psychosomatic day clinic, individual and group sessions took place with the following contents: group psycho-education, explanation of the treatment rationale (group), group and individual exposure therapy with interoceptive exposure, therapist-accompanied and self-managed confrontation with feared situations (individual), cognitive restructuring of dysfunctional, anxious beliefs (group and individual). All the therapists were experienced in CBT and regularly supervised by a certificated supervisor. The LIE took place once, as part of the guided interoceptive exposure.

## Cortisol sampling

Female participants were invited during the luteal phase of their menstrual cycle since for the follicular phase, a higher cortisol level has been repeatedly shown [37]. A total of six saliva samples were collected 15 minutes (-15 min) and one minute before (-1 min) as well as one minute (+1), +10, +20 and +30 minutes after LIE, by way of Salivette swabs (Sarstedt, Nümbrecht, Germany). Samples were then kept frozen at –20 C before being assayed for cortisol, as primary outcome measure of the present study. For preparing the examination and producing a clear supernatant of low viscosity, samples were centrifuged at 3.000 rpm for 5 min. For cortisol analysis, 50 µL were removed using a commercially available immunoassay with chemiluminescence detection [11].

## Heart rate monitoring

The patients were fitted a belt for continuous wireless transmission of heart rate signals with electrocardiogram precision via a polar system (S810, Polar ® Electro GmbH Deutschland). Mean heart rate, as another primary outcome measure of the present study, was calculated as beats per minute over three-minute intervals, during each part of the testing procedure (baseline: 1–3 minutes before LIE, during LIE, 1–3 minutes after LIE). Before LIE, a guided breathing session took place. This included six breaths per minute for a total duration of five minutes (Fig 2).

## Clinical measures

Sociodemographic variables (age, family status, smoking, alcohol use), health status (medical anamnesis, Body Mass Index/ BMI) and medication intake was assessed via a routine medical examination.

**1.)** The state anxiety was a primary outcome measure. It was measured using the state version of the *State Trait Anxiety Inventory (STAI-state/ STAI-s)* before and after LIE [38]. The STAI-state consists of 20 items, measuring the intensity of anxiety as emotional state on a 4-point Likert scale (20 to 80). **2.)** As further primary outcome measure, the affective dimensions of experienced ‚arousal‘, ‚pleasure‘ and ‚dominance‘ were assessed using the *Self-Assessment Manikins*. The latter are non-verbal, nine- point pictorial rating scales and were applied both before and after LIE [39].

The questionnaires for the assessment of symptom severity, as secondary outcome measures, were applied both before and after an intensive cognitive-behavioral treatment at a psychosomatic day clinic. **3.)** The Panic and Agoraphobia Scale (PAS) [40] is a measure of the severity of PD with/ without agoraphobia. It contains 13 items with five-point Likert scales (0 to 3), summed to obtain a total score (range 0–52). The items are grouped in five subscales (panic attacks, agoraphobic avoidance, anticipatory anxiety, disability, worries about health). **4.)** The *Mobility Invento*ry (MI) [41] is a 27-item measurement of agoraphobic avoidance behavior and frequency of panic attacks, according to 26 situations. These are rated on a 5-point scale, both when patients are accompanied and when they are alone. Item scores are

averaged to yield the total score (range 0–4). **5.)** The *Agoraphobic Cognitions Questionnaire (ACQ)* [42] measures fearful cognitions associated with panic attacks and agoraphobia. The 14 items are rated on a 5-point scale and averaged to obtain a mean score (range 0–4). **6.)** The *Body Sensations Questionnaire (BSQ)* [42] measures the intensity of fear associated with particular physical symptoms of arousal. The 17 items are evaluated on a 5-point Likert scale and then averaged (range 0–4). **6.)** The Beck-Depression-Inventory (BDI) [43] was applied in order to assess the intensity of depression. The 21 items are rated on a 4-point scale (0–3) and summed up to a total score (range: 0–63).

### Statistical analysis

Normality of variables was controlled for using the Kolmogorov-Smirnov test. In case of normally distributed and continuous data, groups were compared using analysis of covariance (ANCOVA), controlling for age and sex. The latter was not controlled for when calculating therapy responses, facing the small sample size of n = 13 patients. For categorical variables, Chi-square tests were applied. Mann-Whitney- and Wilcoxon tests were used in case of nonnormally distributed and ordinal data. Self-reported distress (STAI-s, Self-Assessment Manikins), age, salivary cortisol and heart rate data were logarithmized (logarithm with base 10), in order to reduce skewness of data. Baseline group differences in the three Self-Assessment Manikins dimensions and state anxiety (STAI-s) were assessed using Mann-Whitney U-tests. Changes in perceived state anxiety (STAI-s) and Self-Assessment Manikins-dimensions were tested, using Wilcoxon tests. Group differences in the course of salivary cortisol were calculated by means of a 2 (group: PD, healthy controls) by 6 (time: -15, -1, +1, +10, +20, +30 min) ANCOVA for repeated measures, taking into account sex, age and baseline cortisol (-1min) as covariates. The course of heart rate was assessed using a 2 (group: PD, healthy controls) by 4 (baseline 1, baseline 2, stress, recovery) ANCOVA with age and sex as covariates. Greenhouse Geisser corrected values were used to account for violation of the spherecity assumption. The Area under the response curve was calculated with respect to ground (AUCg) (measuring total cortisol output) and increase (AUCi) (measuring change in cortisol regardless of baseline), according to Pruessner, Kirschbaum [44]. The heart rate response was investigated using a delta score (difference between maximum heart rate after LIE and baseline heart rate). Spearman´s rank correlations were performed to examine the association between cortisol secretion (AUCg/ AUCi)/ heart rate response under LIE and the most relevant psychopathological measure (PAS). For this purpose, the PAS total and subscale scores were transferred into percentages of the respective baseline scores (low values representing high therapy success) [6]. Exploratory only, a hierarchical regression analysis was calculated, with the baseline cortisol value (raw value) entered in a first step and AUCg cortisol (raw value) in a second one, using Enter as method. Significance was accepted at $p \leq .05$. Bonferroni-corrections of p-values were applied where appropriate (see Table 2). Data analyses were performed using SPSS v. 28.0.0.0 (SPSS Inc., Chicago, IL, USA). The data file can be retrieved from the Open Science Framework (https://archive.org/details/osf-registrations-s76rw-v1, source https://osf.io/s76rw/, registration DOI 10.17605/OSF.IO/S76RW, identifier DOI 10.17605/OSF.IO/NE4G6).

### Results

30 patients were consecutively assessed for eligibility. Of them, nine patients refused study participation. One patient and one healthy control were excluded, because their cortisol values laid two standard deviations above the mean of the respective group. Another healthy control had to be excluded because of missing values. A total number of n = 20 patients and n = 20 healthy control subjects were examined according to their psychopathological measures,

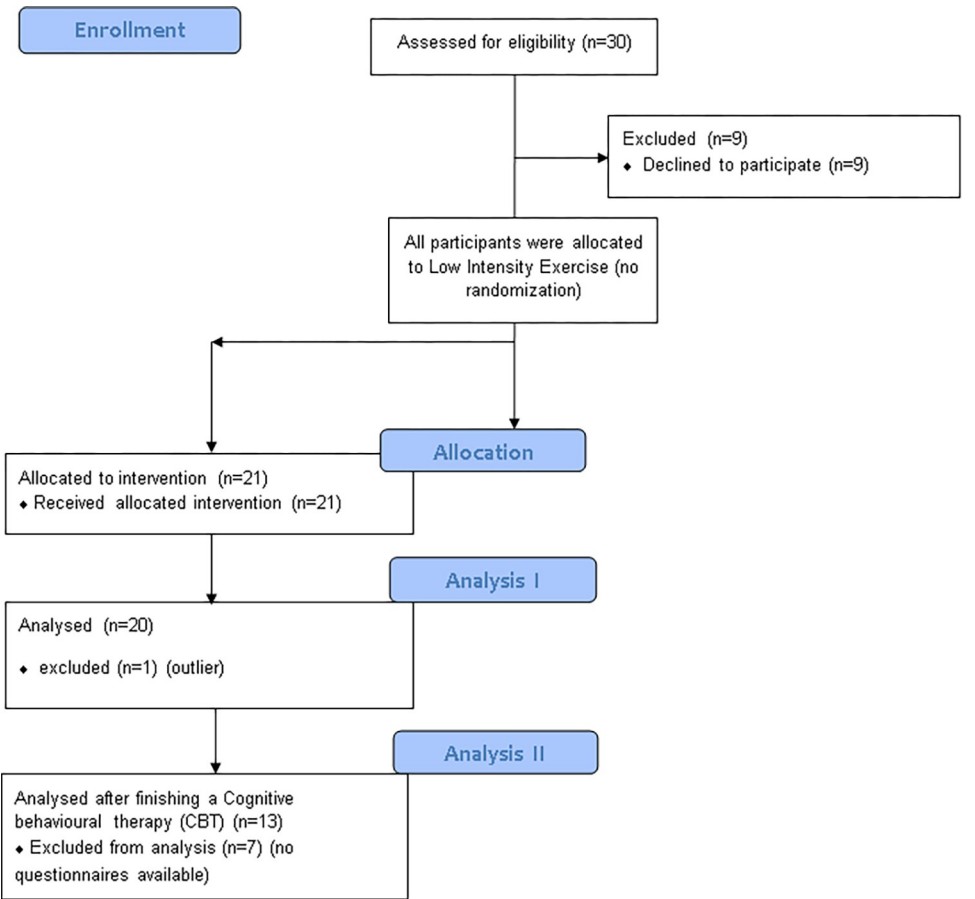

**Fig 1. Flow chart showing study enrollment, allocation and number of patients analysed.**

cortisol secretion pattern and heart rate under LIE (Fig 1). The patients who dropped out did not significantly differ from the final sample with respect to the main sociodemographic and psychopathological characteristics as well as the BMI (see S1 Table). After CBT, a subsample of 13 patients was analysed since data of seven patients was missing (Fig 1).

Of the final sample, a PD with agoraphobia was diagnosed in n = 16 (80%) patients. The remaining n = 4 (20%) patients had a PD without agoraphobia. Median onset of PD was 28 yrs (IQR: 19.8–34.3), median duration 3.0 yrs (IQR: 1.1–12.2). Both groups were statistically well matched with respect to age and sex. Significantly more patients than healthy controls lived in a partnership. No differences were obvious according to most of the basic medical variables (e.g. body mass index, smoking status, use of oral contraceptives, medical illnesses). An overview of sociodemographic and psychopathological data is provided in Table 1.

## Clinical measures: Effect of therapy

Symptoms of PD significantly decreased to 52.0% (mean, SD: 35.6) of the baseline value, as measured by the Panic and Agoraphobia Scale (PAS) ($F_{(1,12)}$ = 24.831, p < .001, partial $\eta^2$ = .674). Likewise, improvement could be shown according to anxiety cognitions (ACQ) ($F_{(1,12)}$ = 9.582, p = .009, $\eta^2$ = .444; median: 65.2%, IQR: 43.8–113.3) and fear of body sensations (BSQ) ($F_{(1,12)}$ = 6.244, p = .028, $\eta^2$ = .342; mean: 79.3%, SD: 41.9).

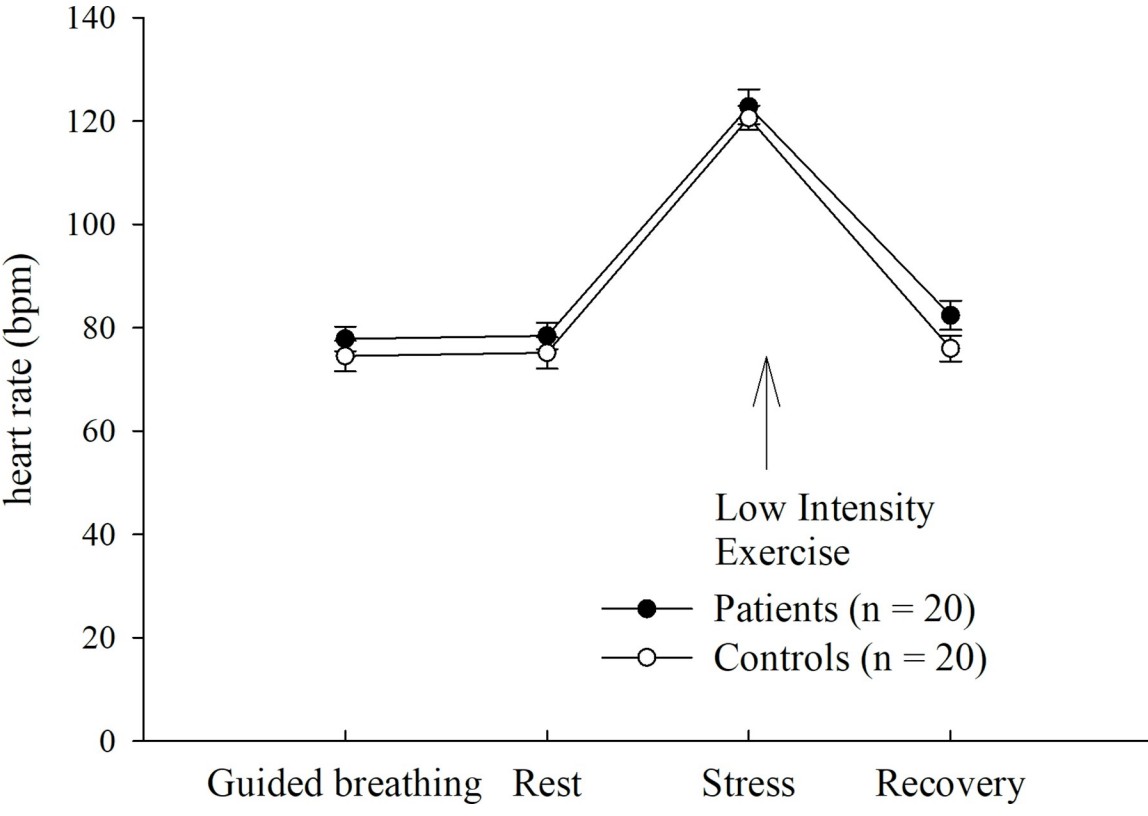

**Fig 2. Heart rate response under Low Intensity Exercise (LIE) in patients with panic disorder and healthy controls. Abbreviations/** see S3 File: bpm, beats per minute; LIE, low intensity exercise.

### State anxiety

Before low intensity exercise, patients showed a tendency towards higher state anxiety (STAI-s) than healthy control subjects (U = 123.000, p = .092). While LIE did not lead to a significant increase of state anxiety in patients, control subjects´ state anxiety did (patients: Z = -.464, p = .643, controls: Z = -2.442, p = .015).

### Self-assessment manikins

Groups did not significantly differ with respect to perceived pleasure, arousal and dominance before LIE (Self-Assessment Manikins pleasure: U = 171.500, p = .783; Self-Assessment Manikins arousal: U = 139.000, p = .319; Self-Assessment Manikins dominance: U = 156.000, p = .458). While LIE did not lead to a significant change of SAM dimensions in patients with PD (Self-Assessment Manikins pleasure: Z = -.577, p = .564, Self-Assessment Manikins arousal: -1.637, p = .102, Self-Assessment Manikins dominance: Z = -1.131, p = .258), controls reported a significant decrease of perceived pleasure (Z = -2.414, p = .016). There was no significant change of arousal and dominance in controls (SAM 1: Z = -1.656, p = .098; SAM 3: Z = -1.190, p = .234).

### LIE heart rate response

With respect to heart rate, there was no significant baseline difference (F(1,35) = .932, p = .342, partial $\eta^2$ = .029). Under LIE, heart rate significantly increased irrespective of group (main

**Table 1. Sociodemographic, medical, psychopathological variables and subjective level of distress of the N = 20 patients with PD with/ without agoraphobia and a sample of healthy controls (N = 20).**

| | Patients with PD with/ without agoraphobia n = 20 | Healthy controls n = 20 | F/ $\chi^2$/ U | P |
|---|---|---|---|---|
| **Sociodemographic variables** | | | | |
| Sex, n (%) | | | 1.600 | .206($\chi^2$) |
| females | 12 (60) | 8 (40) | | |
| males | 8 (40) | 12 (60) | | |
| Age, years median, IQR | 29.8 (22.7–43.3) | 24.1 (22.0–28.9) | 141.000 | .114(U) |
| Familiy status, n (%) | | | 13.389 | .010*($\chi^2$) |
| Single/ no partner | 5 (25) | 9 (45) | | |
| Married | 5 (25) | 0 | | |
| cohabitation (not married) | 10 (50) | 11 (55) | | |
| Education, n (%) | | | 6.933 | .074($\chi^2$) |
| No graduation | 0 (0) | 0 (0) | | |
| Hauptschule/ Realschule (Secondary School leaving certificate) | 9 (45) | 2 (10) | | |
| High School (Abitur) | 7 (35) | 14 (70) | | |
| University | 4 (20) | 4 (20) | | |
| **Medical variables** | | | | |
| Smoking, n (%) | 6 (30) | 6 (30) | .000 | 1.000($\chi^2$) |
| Regular sports, n (%) | 12 (60) | 13 (65) | .107 | .744($\chi^2$) |
| Cycle week, n (%) | | | 1.800 | .407($\chi^2$) |
| 3rd week | 9 (75) | 6 (62.5)[c] | | |
| 4th week | 1 (8.3) | | | |
| menopause | 2 (16.7) | | | |
| Use of oral contraceptives, n (%) | 6 (50) | 3 (37.5) | .303 | .670($\phi$) |
| Body Mass Index (BMI), median (IQR) | 22.2 (20.6–25.1) | 24.5 (21.7–26.5) | 153.000 | .204 (U) |
| **Psychopathological measures** | | | | |
| PAS total score [0–52], median (IQR) | 17.2 (5.9–24.5) | 0 (0–0)[a] | 32.500 | < .001***(U) |
| ACQ total score [0–4] | .9 (.6–1.2) | .5 (.3-.8) | 93.000 | .004**(U) |
| BSQ total score [0–4] | 1.5 (1.0–2.2) | 1.0 (.5–1.4) | 106.000 | .011*(U) |
| MI alone [0–4] | .5 (.2–1.7) | .1 (.0-.3) | 96.500 | .005**(U) |
| MI accompanied [0–4] | .4 (.1–1.1) | .0 (.0-.1) | 70.000 | < .001***(U) |
| BDI [0–63] | 7.5 (5.3–12.8) | 5.0 (1.3–7.0)[d] | 80.000 | .011*(U) |
| **Subjective level of distress** | | | | |
| Stai-state before Low Intensity Exercise (LIE) [20–80] | 34.7 (31.6–42.0)[b] | 32.0 (31.0–36.0)[b] | 123.000 | .092(U) |
| Self-Assessment Manikins pleasure before LIE [1–9] | 7.0 (6.0–7.0)[b] | 7.0 (6.0–7.0)[b] | 171.500 | .783(U) |
| Self-Assessment Manikins arousal before LIE [1–9] | 3.0 (2.0–5.0)[c] | 3.0 (1.0–4.0)[b] | 139.000 | .319(U) |
| Self-Assessment Manikins dominance before LIE [1–9] | 6.0 (5.0–7.0)[b] | 5.0 (5.0–6.0)[b] | 156.000 | .458(U) |
| Stai-state after LIE [20–80] | 34.0 (28.0–51.0)[b] | 36.0 (33.0–38.0)[b] | 173.000 | .826(U) |
| Self-Assessment Manikins pleasure after LIE [1–9] | 7.0 (5.0–7.0)[b] | 7.0 (6.0–8.0)[b] | 137.000 | .187(U) |
| Self-Assessment Manikins arousal after LIE [1–9] | 5.0 (3.0–5.0)[b] | 3.0 (1.0–5.0)[b] | 150.500 | .368(U) |
| Self-Assessment Manikins dominance after LIE [1–9] | 7.0 (5.0–7.0)[b] | 6.0 (5.0–7.0)[b] | 154.000 | .424(U) |

*p $\leq$ .05

** p $\leq$ .01

*** p $\leq$ .001; $\phi$ = Fisher´s exact test

[a] n = 3 missing values

[b] n = 1 missing value

[c] n = 2 missing values

[d] n = 4 missing values

**Abbreviations**/ see S3 File: ACQ, Agoraphobic Cognitions Questionnaire; BDI, Beck-Depression-Inventory; BMI, Body Mass Index; BSQ, Body Sensations Questionnaire; LIE, Low Intensity Exercise; MI, Mobility Inventory; PAS, Panic and Agoraphobia Scale; Stai-state, State Trait Anxiety Inventory

**Table 2. Spearman´s correlations between cortisol response under Low Intensity Exercise (LIE) (AUCg/ AUCi) and the PAS score at T1 (n = 20), T2 (n = 13) and percent change in PAS from T1 to T2.**

| | T1 | | T2 | |
| --- | --- | --- | --- | --- |
| | AUCg | AUCi | AUCg | AUCi |
| **PAS total** | -.225 (.340) | **-.495** (.026*) | -.455 (.119) | -.493 (.087) |
| **Panic attacks (PA)** | .130 (.585) | -.251 (.286) | -.369 (.215) | -.459 (.115) |
| **Agoraphobic avoidance (AV)** | -.185 (.435) | **-.607** (.005**) | -.203 (.507) | **-.588** (.035*) |
| **Anticipatory anxiety (AA)** | -.283 (.227) | -.287 (.220) | -.304 (.313) | **-.585** (.036*) |
| **Disability (D)** | -.309 (.185) | **-.446** (.049*) | -.238 (.434) | -.316 (.293) |
| **Health concerns (HC)** | -.239 (.310) | -.174 (.463) | **-.677** (.011*) | -.090 (.770) |
| **PAS total percentage of baseline (pob)** | | | -.496 (.085) | -.298 (.324) |
| **PA (pob)** | | | -.463 (.111) | -.313 (.298) |
| **AV(pob)** | | | -.294 (.329) | -.380 (.200) |
| **AA (pob)** | | | -.387 (.192) | **-.554** (.050*) |
| **D (pob)** | | | -.117 (.702) | -.193 (.528) |
| **HC (pob)** | | | **-.738** (.004**)# | .022 (.942) |

*p ≤ .05

** p ≤ .01

*** p ≤ .001

# significant at Bonferroni-corrected p-value: p ≤ .01

**Abbreviations/** see S3 File: AA, Anticipatory Anxiety; AUCg/i, Area under the Curve with respect to ground/ increase; AV, Agoraphobic Avoidance; D, Disability; HC, Health Concerns; PA, Panic Attacks; PAS, Panic and Agoraphobia Scale; POB, percentage of baseline

effect of time: F(3,90) = 4.391, p = .018, partial $\eta^2$ = .128) (see Fig 2). There were neither a significant main effect of group (F(1,30) = 1.481, p = .233, partial $\eta^2$ = .047) nor a significant group by time interaction (F(3,90) = .904, p = .406, partial $\eta^2$ = .029). Also, the impact of sex on heart rate response could be ruled out (main effect of sex: F(1,30) = .424, p = .520, partial $\eta^2$ = .014, interaction: time x sex: F(3,90) = 1.273, p = .287, partial $\eta^2$ = .041).

## LIE cortisol response

Both groups started from a similar baseline cortisol value (-1min) (F(1,39) = 1.471, p = .233, partial $\eta^2$ = .040). By tendency only, patients with PD showed a slightly higher baseline value than healthy controls (6.4 nmol/ l, IQR 4.6–10.3 vs. 5.6 nmol/l, IQR 3.3–7.1). LIE led to a significant change of cortisol secretion over time, in both groups (main effect of time: F(5, 170) = 3.541, p = .022, partial $\eta^2$ = .094). Descriptively only: while patients showed their highest cortisol value shortly before the LIE, controls reached their peak at 20 minutes following LIE. Separate analyses within subgroups revealed that the cortisol value slightly decreased over time, especially in male patients (patients, interaction time x sex: F(5,80) = 3.450, p = .038, partial $\eta^2$ = .177; controls, interaction time x sex: F(5,80) = .868, p = .441, partial $\eta^2$ = .051), whereas no significant change occurred in healthy controls (see Figs 3 and 4). There were neither a significant main effect of group (F(1, 34) = .200, p = .658, partial $\eta^2$ = .006), nor a significant group by time interaction (F(5, 170) = .926, p = .423, partial $\eta^2$ = .027).

## Association between heart rate response/ cortisol and therapeutic outcome

With respect to the heart rate response under LIE, no significant correlation could be found with the psychopathology before the intensive CBT (Spearman´s rho heart rate delta T1 x PAS

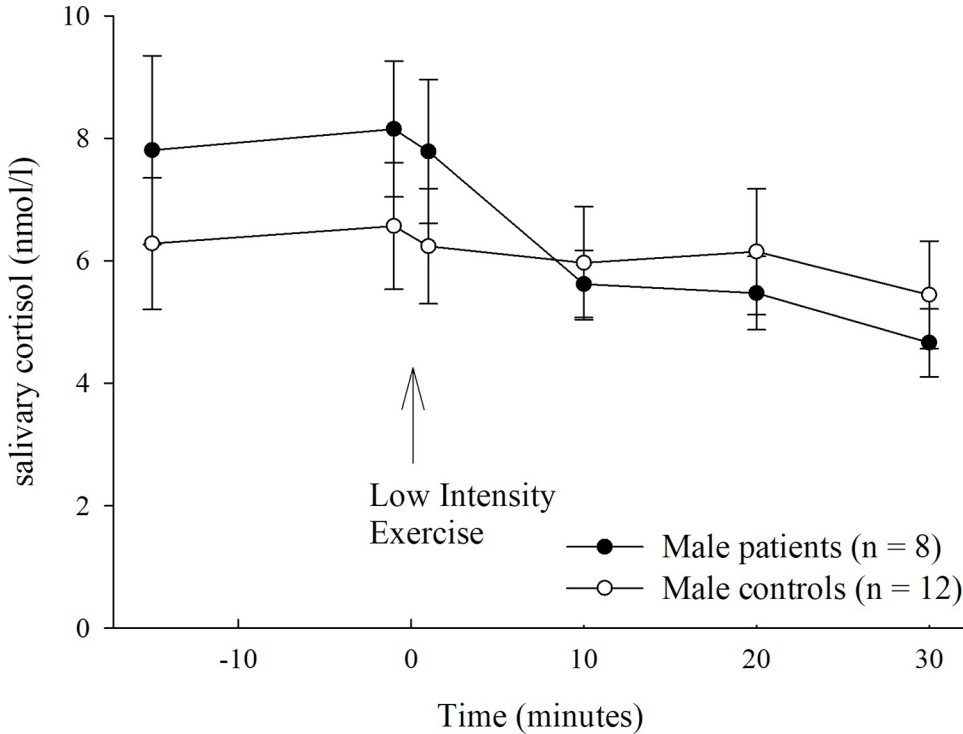

**Fig 3. Cortisol response under Low Intensity Exercise (LIE) in male patients with panic disorder and healthy controls.** Abbreviations/ see S3 File: LIE, low intensity exercise.

T1 = -.090, p = .723). The lowest heart rate response under LIE was associated with the least therapeutic outcome (percent change in PAS from T1 to T2), after intensive CBT (higher percentage of baseline). This was significant for the PAS-subscale 'panic attacks (PA)', only (Spearman´s rho Delta heart rate x PA percent baseline T1/ T2 = -.724, p = .008).

A significantly negative correlation of the cortisol response under LIE (AUCi) with the psychopathology before intensive CBT (PAS) could be shown in the patient group (Spearman´s rho AUCi T1 x PAS T1 = -.495, p = .026). Patients with the highest psychopathology presented the lowest cortisol response, due to the physiological stressor. In particular, the subscale ‚agoraphobic avoidance (AV)' before CBT was negatively correlated with the cortisol response (Spearman´s rho AUCi T1 x AV T1 = -.607, p = .005, significant at Bonferroni-corrected p-value p ≤ .01) (see Table 2). After CBT, there was a significantly negative correlation of the cortisol response under LIE, with the PAS-subscales ‚agoraphobic avoidance', anticipatory anxiety' and ‚health concerns'. By tendency, the lowest cortisol response under LIE was associated with the least therapeutic outcome (percent change in PAS from T1 to T2) after intensive CBT (higher percentage of baseline). This result was significant for the subscales ‚anticipatory anxiety (AA)' (Spearman´s rho AUCi T1 x AA percent baseline T1/ T2 = -.554, p = .050) and ‚health concerns (HC)' (Spearman´s rho AUCg T1 x HC percent baseline T1/ T2 = -.738, p = .004, significant at Bonferroni-corrected p-value p ≤ .01) (see Table 2, Fig 5). The hierarchical linear regression model including baseline cortisol and the HPA axis activity (AUCg) explained 54.7% of the data variance (F(2,12) = 8.255, p = .008). However, the impact of the AUCg (cortisol) on the course of 'health concerns' just failed to reach significance (β = -.962, T = -2.208, p = .052, 95% CI -.539, .002).

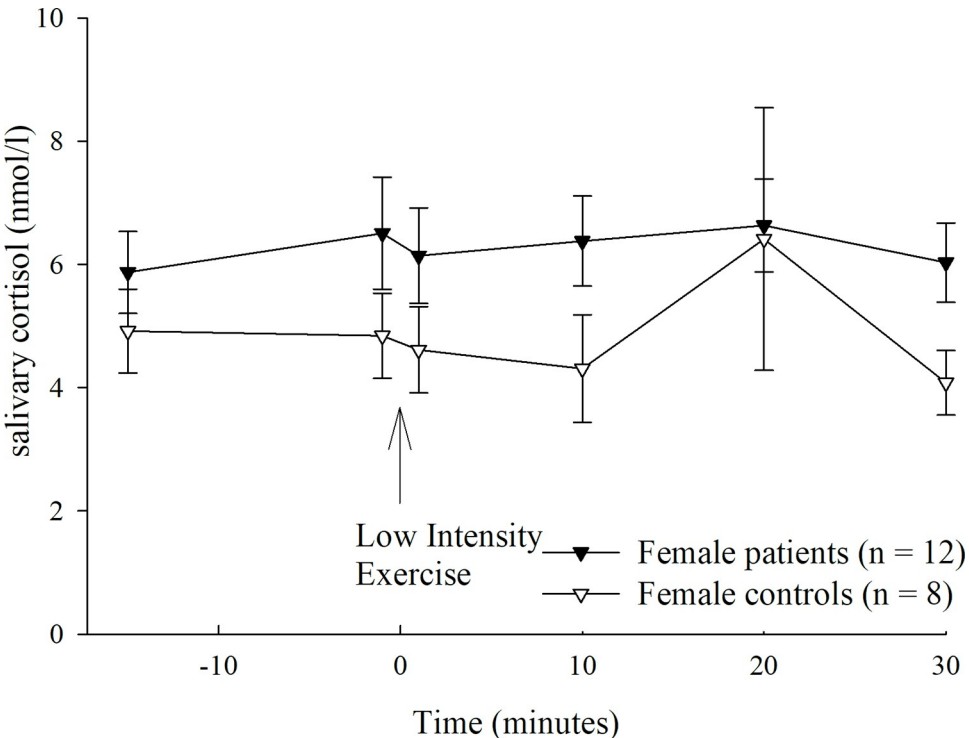

**Fig 4. Cortisol response under Low Intensity Exercise (LIE) in female patients with panic disorder and healthy controls. Abbreviations/** see S3 File: LIE, low intensity exercise.

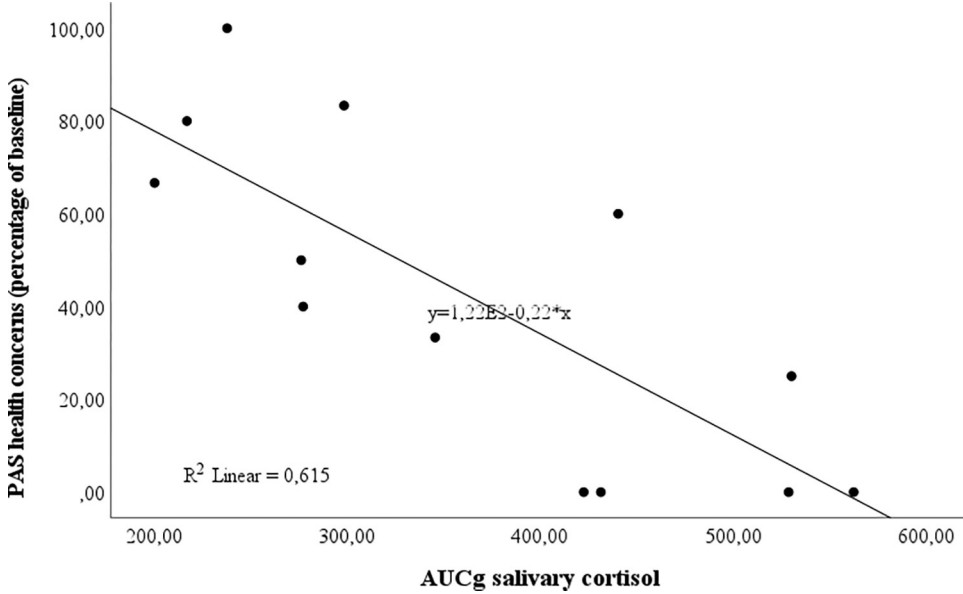

**Fig 5. Correlation between salivary cortisol (AUCg) and the PAS subscale health concerns.** The lower the cortisol response under Low Intensity Exercise (LIE), the lower the improvement (higher percentage of baseline) in PAS health concerns under intensive cognitive-behavioral therapy. **Abbreviations/** S3 File: AUCg, Area under the Curve with respect to ground; LIE, low intensity exercise; PAS, Panic and Agoraphobia Scale.

## Discussion

### General discussion

The present study investigated the induction of anxiety and course of stress parameters (cortisol and heart rate response) under stimulation with low intensity exercise (LIE), which was realized as interoceptive exposure, after the beginning of an intensive cognitive-behavioral therapy in patients with PD with/ without agoraphobia. We were interested whether a LIE displays an appropriate and effective interoceptive stressor, leading to a sufficient stress response. As a further study aim, we were interested whether the cortisol/ heart rate responses are associated with therapeutic outcome in these patients. The findings could be of clinical relevance, since an adequate stress response is necessary in order to effectively cope with stressors. Additionally, it may lead to fear desensitization and prevent adverse physiological long-term effects in PD [10]. Deficits in mounting and ending an adequate stress response can lead to a maladaptation under stress, and increase the probability for the development of psychopathology as well as chronic-inflammatory diseases [45]. Hence, the results may help to improve exposure therapy in these patients and explain therapy non-response [6, 26].

The results confirm significant SAM-responses under LIE. This finding is in line with reports on more intensive exercise conditions (e.g. bicycle ergometry) by Kirschbaum, Strasburger [28], Skoluda, Strahler [27] and Allgrove, Gomes [46], showing a significant sympathetic nervous system activity with a marked increase of heart rate. However, with respect to HPA-axis response, no significant increase of cortisol secretion was obvious. Generally, in our present study, only a minor number of patients and controls showed a significant increase of cortisol levels of at least 2.5nmol/l, as suggested by Van Cauter and Refetoff [47] (patients: n = 2/ 10%, controls: n = 4/ 20%). Also, using a laxer cut-off for cortisol response (1.5nmol/l) [48], still brought about a quitely low number of responders, in either group (patients: n = 2/ 20%, controls: n = 5/ 25%). In line, also Kirschbaum, Strasburger [28], Allgrove, Gomes [46] and Prehn-Kristensen, Wiesner [35] could not find any significant cortisol responses, even when a more intensive paradigm such as bicycle ergometry was used. This in turn is in contrast to the findings by Skoluda, Strahler [27]. The contradicting findings may be explained by differences in the design of the exercise task (different intensities, different durations) and the characteristics of the sample (e.g. sample size, habitual smokers vs. non-smokers, training status of participants). Circulating cortisol is dependent on the intensity of exercise, with elevations occurring following intensities above approximately 60% of the maximal oxygen uptake ($V_{O2max}$) [49]. Former protocols applied more intensive stimulations, cycling at a higher level, requiring very strong effort, which corresponds to at least 80% of self-rated maximal effort, using the Borg Rating Perceived Exertion (RPE) scale [27]. Or workload was continuously increased by 50 W, every minute until exhaustion [28]. Following, LIE did not lead to a significant cortisol response in the present study, because of the low intensity of the interoceptive stimulus applied. Usually, cortisol elevations occur under intensities above approximately 60% $VO_{2max}$ [46]. Thus, future studies should control maximal oxygen uptake, in order to ascertain a proper cortisol response. Above, exercise, when applied as interoceptive exposure during cognitive-behavioral therapy in PD, should therefore be conceptualized as stimulation of high-intensity and continuous exposure, in order to obtain an anxiety induction as well as a significant HPA-axis response. Studies could show that high-level endurance trainings, adjuvant to CBT, had a decelerating effect on the HPA-system, which underlines the anxiolytic effect of exercise for anxiety disorders [50–52]. Above, besides reducing global anxiety sensitivity, only high-intensity exercises (90% of $HR_{max}$) reduced fear of anxiety-related bodily sensations [53] as well as panic reactions to panicogenic agents [54]. Underlining this finding, a recent meta-analysis showed that high self-reported physical activity reduces the risk of developing anxiety

[55]. However, the research on the effect of exercise on anxiety symptoms still remains controversial [for systematic reviews: 56]. Thus, in order to gain more insight into the antipanic mechanisms of physical exercise, future studies should design high-intensity exercise stimulation, use structured training protocols with careful documentation of the intensity, duration and frequency of the training and investigate to what extent the anxiety response as well as SAM/ HPA-axis activity changes. Not least, future studies should compare the impact of different intensities of interoceptive exposures, like light vs. moderate or intensive exercise, e.g. using a randomized controlled trial.

Another explanation for the lack of cortisol increase under LIE may be the avoidance of exercise in patients with PD. In line, we found a negative association of low cortisol response with increased agoraphobic avoidance. Above, former studies showed dysfunctional, defensive mobilization, especially in highly anxiety sensitive patients [23]. This should be investigated in future studies, comparing patients with low vs. high anxiety sensitivity.

In the present study, no anxiety induction took place. It is conceivable that the stressor applied was too little intensive, although patients with PD usually report a higher anxiety sensitivity and react more anxiously, even when faced with a stimulus of low intensity. In line, Skoluda, Strahler [27] and Lattari, Budde [57] showed that a more intensive physical task (bicycle ergometry) was appraised as being more threatening/ distressing than the rest condition.

Patients started from an already heightened baseline state anxiety, compared with healthy controls, precluding an additional increase. In line, our results rather suggest a steady decrease in cortisol secretion under low intensity exercise, particularly in male patients. This is consistent with evidence corroborating rather a decrease than increase under a low intensity exercise [29]. The latter could not be proven for the heart rate response. Increased baseline cortisol values have been already shown by other reports, using differing stress protocols [6, 7, 9]. The increased baseline cortisol value may be associated with a heightened sensitivity to contextual cues, such as novelty [9]. Patients seem to react more sensitive to the whole testing procedure, although more than two third of them are familiar with regular sport activities. Future studies should assess in more detail which kind of sports they are doing regularly. Moreover, the HPA-axis has been shown to be especially responsive to certain kinds of stressors, like social-evaluative stress, unpredictability and uncontrollability [58]. Thus, it is imaginable that patients react more pronounced than healthy controls to the possibly negative social evaluation of the test procedure. This matches the phenomenology and nosology of PD with/ without agoraphobia and its overlap with social anxiety [2]. With respect to future research, it is advisable that the baseline time should be extended to more than 15 minutes, in order to enable the patients to more properly adapt to the test procedure and to launch a significant response of the HPA-axis [30]. Moreover, there is a heterogeneous literature on how well salivary cortisol concentrations represent HPA-axis activity, assuming different modulators [for a systematic review: 59]. For instance, we found a decline rather than increase in the cortisol response, particularly in male than in female patients and controls, respectively. Consistently, gender differences with overall lower cortisol responses in female than male patients have been already shown in former research, whereby an estradiol induced alteration of the glucocorticoid-binding proteins is supposed [5, 60].

As one major finding, we showed significantly negative correlations between the cortisol secretion, and partially the heart rate response under LIE, and psychopathology. A lower response of the HPA-axis/ heart rate was associated with less therapy response, after a 5-week cognitive-behavioral therapy. Similar findings have been already described by Siegmund et al. [6], showing that the lowest cortisol responses during exposure therapy was associated with the highest psychopathology at baseline and least improvement by therapy. In line, findings by Wichmann et al. [26] confirm the inverse relationship between cortisol response to a stressor

and agoraphobic avoidance behaviour after psychotherapy. Finally, results by Meuret et al. emphasize a higher absolute cortisol level during exposures as moderator for clinical improvement (avoidance behavior, threat appraisal, perceived control) [61]. In our study, the above mentioned associations were present for the PAS-subscales 'panic attacks', ,anticipatory anxiety', ,agoraphobic avoidance' and ,health concerns'. It is conceivable, that patients with a higher intensity of these symptoms, avoid psychophysiological arousal under interoceptive exposure such as low intensity exercise—which is mirrored by an absent increase in perceived arousal due to LIE. While these patients showed an increase in the activity of the sympathetic nervous system, neither arousal nor state anxiety increased. Thus, these patients presented some kind of dissociation between the SAM- and HPA-axis stress response, assumably making extinction processes more difficult. An increased excitability of hippocampal neurons by norepinephrine is supposed. Consequently, the retrieval of panic-related information may be triggered, and the consolidation of new, non-fearful memory may be inhibited [62–64]. Following, the kind of exercise as used in the present study is not suitable in order to track memory extinction processes. Our results are in line with other research (see systematic review) [65]. However, the findings are still equivocal of whether higher cortisol concentrations during exposure therapy are linked with better treatment outcomes.

Our present results should be critically reviewed in the context of methodological strengths and shortcomings. We used a quite homogeneous sample of patients and ascertained the diagnosis of a PD via a the SCID-I [32]. Above, we applied a quite stringent stress protocol of ecological validity, as part of the cognitive-behavioral therapy during a semi-residential care. Patients were free of any comorbid psychological disorder and did not receive psychopharmaceutical drug treatment.

## Limitations of the study

The present results should be critically reviewed in the context of methodological shortcoming. The study was originally planned as part of another study, collecting stress-related body odors under physical and psychosocial stress [31] (for more details see the study protocol in the supplementary material). The present study may be rather evaluated to be a pilot or exploratory study. Following, the results should be interpreted with caution and as preliminary. Only a small sample size was enrolled. Nevertheless, when a large effect size was supposed (calculating ANCOVAs), still a test power (1-beta error probability) of about 34% could be reached. In future replication studies, a sample size of at least 43 patients per group, is necessary in order to reach a statistically sufficient power of 80%. The present sample was not exactly equally paralleled with respect to age and sex. P-value adjustment for multiple testing was not applied in all statistical analyses (only where from our point of view was appropriate, see Table 2). Above, we neither applied increasing intensities of the low intensity exercise task, comparing different intensity conditions, nor controlled for the maximal oxygen consumption as indicator of the endurance capacity. Also, we did not examine the stress parameters before the cognitive-behavioral treatment and did not realize a follow-up several months thereafter [6]. Future investigations should be thoroughly designed, double-blind Randomized Controlled Trials (RCT), comparing low with moderate/ high intensity of exercise and should examine differing subgroups of patients (e.g. with fear of suffocation vs. fear of cardiological events, high vs. low anxiety sensitivity).

## Conclusions

In the present study, patients with PD undergoing ten minutes of light intensity exercise, showed divergent stress responses. While heart rate increased, cortisol decreased, mainly in

male patients. A lower reactivity of the HPA-axis was associated with a higher panic-related psychopathology and a lower treatment outcome, following an intensive cognitive-behavioral psychotherapy. The results allude that an insufficient HPA-axis response may go along with disturbed extinction-based learning. Moreover, in order to be effective, physical/ interoceptive stimulation should be designed as more intense and potent stressor, even in anxiety sensitive disorders such as the PD.

## Supporting information

**S1 Checklist. TREND statement checklist.**
(PDF)

**S1 Table. Baseline characteristics (sociodemographic and psychopathological variables, BMI) of the N = 20 patients with PD with/ without agoraphobia and the sample of dropped out patients (N = 10).**
(DOCX)

**S1 File. Study protocol- English.**
(DOCX)

**S2 File. Study protocol- German.**
(DOCX)

**S3 File. List of abbreviations.**
(DOCX)

## Acknowledgments

The authors would like to thank Prof. Katja Petrowski from the University Medical Center of the Johannes Gutenberg-University Mainz, Department of Psychosomatic Medicine and Psychotherapy, Medical Psychology and Medical Sociology, for supervising the SCID assessments. We would also like to thank the laboratory from Prof. Clemens Kirschbaum, for analyzing the cortisol samples of the present study and Dr. Deborah Löllgen for supporting the heart rate analyses.

## Author Contributions

**Conceptualization:** Gloria-Beatrice Wintermann, Susann Steudte-Schmiedgen.

**Formal analysis:** Gloria-Beatrice Wintermann.

**Funding acquisition:** Gloria-Beatrice Wintermann.

**Investigation:** Gloria-Beatrice Wintermann.

**Methodology:** Gloria-Beatrice Wintermann.

**Project administration:** Gloria-Beatrice Wintermann.

**Resources:** Gloria-Beatrice Wintermann.

**Software:** Gloria-Beatrice Wintermann.

**Supervision:** Gloria-Beatrice Wintermann.

**Validation:** Kerstin Weidner.

**Visualization:** Gloria-Beatrice Wintermann.

**Writing – original draft:** Gloria-Beatrice Wintermann, Susann Steudte-Schmiedgen.

**Writing – review & editing:** René Noack, Susann Steudte-Schmiedgen, Kerstin Weidner.

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
