## [Decision Letter · Decision Letter 0]

4 May 2022

PONE-D-22-01757Cortisol response under low intensity exercise during cognitive-behavioral therapy is associated with therapy outcome in Panic Disorder – an exploratory studyPLOS ONE

Dear Dr. Wintermann,

Thank you for submitting your manuscript to PLOS ONE. After careful consideration, we feel that it has merit but does not fully meet PLOS ONE’s publication criteria as it currently stands. Therefore, we invite you to submit a revised version of the manuscript that addresses the points raised during the review process.

We look forward to receiving your revised manuscript.

Kind regards,

Walid Kamal Abdelbasset, Ph.D.

Academic Editor

PLOS ONE

Journal Requirements:

The present study was funded by the Robert-Pfleger-Stiftung. GBW received the award.

Reviewers' comments:

Reviewer's Responses to Questions

**Comments to the Author**

1. Is the manuscript technically sound, and do the data support the conclusions?

Reviewer #1: Partly

Reviewer #2: Yes

2. Has the statistical analysis been performed appropriately and rigorously? 

Reviewer #1: Yes

Reviewer #2: I Don't Know

3. Have the authors made all data underlying the findings in their manuscript fully available?

Reviewer #1: Yes

Reviewer #2: No

4. Is the manuscript presented in an intelligible fashion and written in standard English?

Reviewer #1: Yes

Reviewer #2: Yes

5. Review Comments to the Author

Reviewer #1: The statistical analysis of the study was well thought out and the ANCOVA approach was executed with the limited sample size. Obviously a larger study is needed with up front power calculations and specifically stated objectives. The authors do note that future studies should compare the impact of different intensities of interoceptive exposures, like light vs. moderate or intensive exercise, e.g. using a randomized controlled trial.

The investigators gave an honest assessment of the limitations of this study and noted that the present results should be critically reviewed in the context of methodological shortcomings. The sample was not exactly equally paralleled with respect to age and sex. The sample size was quite small with little power, although they do note that a test power of about 34% could be reached when a medium effect size of .25 was supposed (when ANCOVAs were calculated). The present study obviously may be rather evaluated to be a pilot or exploratory study as they note. The descriptive presentation is certainly informative. The investigators should also note with the multiple statistical tests being performed in this limited sample that no adjustment to the p-value has been made for multiple testing and thus the p-values should be interpreted with caution.

Reviewer #2: The study titled (Cortisol response under low intensity exercise during cognitive-behavioral therapy is

associated with therapy outcome in Panic Disorder – an exploratory study) is a clinically important one and nicely presented. I have some recommendations:

1- The title: should be "therapeutic" not "therapy"

2- Methods in general lack references in all parts, this is a point of weakness

3- Check if unnecessary abbreviations are not needed

6. PLOS authors have the option to publish the peer review history of their article (what does this mean?). If published, this will include your full peer review and any attached files.

Reviewer #1: No

Reviewer #2: **Yes: **Sawsan Zaitone

---

## [Author Response · Author response to Decision Letter 0]

23 Jun 2022

PONE-D-22-01757

Cortisol response under low intensity exercise during cognitive-behavioral therapy is associated with therapy outcome in Panic Disorder – an exploratory study

PLOS ONE

Dear Dr. Walid Kamal Abdelbasset , Dear editors,

thank you for your interest in our manuscript entitled „Cortisol response under low intensity exercise during cognitive-behavioral therapy is associated with therapy outcome in panic disorder – an exploratory study“. 

We have adhered to your and the reviewers´ inquiries and revised our manuscript accordingly. Please find our point-by-point replies (marked yellow) in the following rebuttel letter. All further changes in the manuscript are coloured in yellow as well.

Please do not hesitate to contact me in case of further inquiries.

Sincerely Yours,

Gloria Wintermann (on behalf of all co-authors)

Dear Dr. Wintermann,

Thank you for submitting your manuscript to PLOS ONE. After careful consideration, we feel that it has merit but does not fully meet PLOS ONE’s publication criteria as it currently stands. Therefore, we invite you to submit a revised version of the manuscript that addresses the points raised during the review process.

We have attached the document 'Response to Reviewers' accordingly.

We have attached the document 'Revised Manuscript with Track Changes‘ according to your request.

We have attached the document 'Manuscript‘ according to your request.

We have decided to allow public access to the study protocol and anonymous research data using the Open Science Framework (OSF) https://osf.io/.

Access is available using the link: https://osf.io/s76rw/

and https://archive.org/details/osf-registrations-s76rw-v1

The OSF registration doi is the following: 10.17605/OSF.IO/S76RW

We look forward to receiving your revised manuscript.

Kind regards,

Walid Kamal Abdelbasset, Ph.D.

Academic Editor

PLOS ONE

Journal Requirements:

Thank you for this hint. We have adapted the file naming accordingly- both in the text and in the attachments.

The present study was funded by the Robert-Pfleger-Stiftung. GBW received the award.

Please at the financial disclosure as the following statement:

Disclosure

„The author reports no conflicts of interest in this work. The present study was funded by the Robert-Pfleger-Stiftung. GBW received the award. The funders had no role in study design, data collection and analysis, decision to publish, or preparation of the manuscript.“

The corresponding author Gloria-Beatrice Wintermann is affiliated with the chosen institute.

Reviewers' comments:

Reviewer's Responses to Questions

Comments to the Author

1. Is the manuscript technically sound, and do the data support the conclusions?

Reviewer #1: Partly

Reviewer #2: Yes

We have now deposited raw data and supporting information such as the study protocol to a public repository. This will allow to better understand the study design and reproduce the study results.

2. Has the statistical analysis been performed appropriately and rigorously?

Reviewer #1: Yes

Reviewer #2: I Don't Know

3. Have the authors made all data underlying the findings in their manuscript fully available?

Reviewer #1: Yes

Reviewer #2: No

The study data and further supporting information (the study protocol) are available at a public repository called Open Science Framework (OSF) (see above).

The relevant links are the following:

https://osf.io/.

https://osf.io/s76rw/

https://archive.org/details/osf-registrations-s76rw-v1

10.17605/OSF.IO/S76RW

All further information needed are available upon request.

4. Is the manuscript presented in an intelligible fashion and written in standard English?

Reviewer #1: Yes

Reviewer #2: Yes

5. Review Comments to the Author

Reviewer #1: The statistical analysis of the study was well thought out and the ANCOVA approach was executed with the limited sample size. Obviously a larger study is needed with up front power calculations and specifically stated objectives. The authors do note that future studies should compare the impact of different intensities of interoceptive exposures, like light vs. moderate or intensive exercise, e.g. using a randomized controlled trial.

In lines 147-151 we stated the sample size calculation. Study objectives are included on page 4, lines 91-92, page 5, lines 93-94. We also stated that the present study was realized exploratory a spart of another study (see page 5, lines 101-103).

The investigators gave an honest assessment of the limitations of this study and noted that the present results should be critically reviewed in the context of methodological shortcomings. The sample was not exactly equally paralleled with respect to age and sex. The sample size was quite small with little power, although they do note that a test power of about 34% could be reached when a medium effect size of .25 was supposed (when ANCOVAs were calculated). The present study obviously may be rather evaluated to be a pilot or exploratory study as they note. The descriptive presentation is certainly informative. The investigators should also note with the multiple statistical tests being performed in this limited sample that no adjustment to the p-value has been made for multiple testing and thus the p-values should be interpreted with caution.

We included a statement that Bonferroni-corrections of p-values were applied where appropriate (see page 11, line 248/ 249). Significance at Bonferroni-adjusted p-value was indicated in the footnote of Table 2 (page 16).

Reviewer #2: The study titled (Cortisol response under low intensity exercise during cognitive-behavioral therapy is

associated with therapy outcome in Panic Disorder – an exploratory study) is a clinically important one and nicely presented. I have some recommendations:

1- The title: should be "therapeutic" not "therapy"

We have decided to let the title as it is because the term ‚cognitive-behavioral therapy‘ is well-known by the scientific community in this field. We would be glad if you agreed.

2- Methods in general lack references in all parts, this is a point of weakness

Thank you for this hint. We have added lacking references (e.g. for the SCID interview, DSM-IV) and referenced studies the methods applied in the present study were based on (e.g. page 7, lines 141, 143).

3- Check if unnecessary abbreviations are not needed.

For economic reasons, we decided to use the abbreviations because. After introducing them, they were well-defined for the readership (e.g. page 7, line 143). We have prepared a list of abbreviations and included it in the supporting information for better understanding the abbreviations (see supporting information S5 File). Furthermore, we omitted any confusing abbreviations, e.g. SAM, which we have formerly used for both Sympathetic-Adreno-Medullar and Self-Assessment Manikins. In the revised version we decided to use the full-term Self-Assessment Manikins (e.g. see Table 1).

6. PLOS authors have the option to publish the peer review history of their article (what does this mean?). If published, this will include your full peer review and any attached files.

Do you want your identity to be public for this peer review? For information about this choice, including consent withdrawal, please see our Privacy Policy.

Reviewer #1: No

Reviewer #2: Yes: Sawsan Zaitone

Figures have been checked according to the tool you suggested (https://pacev2.apexcovantage.com/ ).

---

## [Decision Letter · Decision Letter 1]

4 Jul 2022

PONE-D-22-01757R1Cortisol response under low intensity exercise during cognitive-behavioral therapy is associated with therapy outcome in panic disorder – an exploratory studyPLOS ONE

Dear Dr. Wintermann,

Thank you for submitting your manuscript to PLOS ONE. After careful consideration, we feel that it has merit but does not fully meet PLOS ONE’s publication criteria as it currently stands. Therefore, we invite you to submit a revised version of the manuscript that addresses the points raised during the review process.

We look forward to receiving your revised manuscript.

Kind regards,

Walid Kamal Abdelbasset, Ph.D.

Academic Editor

PLOS ONE

Journal Requirements:

Reviewers' comments:

Reviewer's Responses to Questions

**Comments to the Author**

1. If the authors have adequately addressed your comments raised in a previous round of review and you feel that this manuscript is now acceptable for publication, you may indicate that here to bypass the “Comments to the Author” section, enter your conflict of interest statement in the “Confidential to Editor” section, and submit your "Accept" recommendation.

Reviewer #1: All comments have been addressed

Reviewer #2: (No Response)

2. Is the manuscript technically sound, and do the data support the conclusions?

Reviewer #1: (No Response)

Reviewer #2: Yes

3. Has the statistical analysis been performed appropriately and rigorously? 

Reviewer #1: (No Response)

Reviewer #2: Yes

4. Have the authors made all data underlying the findings in their manuscript fully available?

Reviewer #1: (No Response)

Reviewer #2: No

5. Is the manuscript presented in an intelligible fashion and written in standard English?

Reviewer #1: (No Response)

Reviewer #2: Yes

6. Review Comments to the Author

Reviewer #1: (No Response)

Reviewer #2: The revised version of paper titled (Cortisol response under low intensity exercise during cognitive-behavioral therapy is

associated with therapy outcome in panic disorder – an exploratory study) is improved compared to the original submission.

I find it necessary that authors change (therapy outcome) int he title to (therapeutic outcome)

7. PLOS authors have the option to publish the peer review history of their article (what does this mean?). If published, this will include your full peer review and any attached files.

Reviewer #1: No

Reviewer #2: No

---

## [Author Response · Author response to Decision Letter 1]

4 Jul 2022

Response: We have now published the English study protocol on protocols.io with the following DOI http://dx.doi.org/10.17504/protocols.io.eq2lynbervx9/v1. A sentence regarding the publication of the study protocol is included in the revised manuscript, as suggested by PlosOne (see page 6, lines 121-122).

Dear Dr. Walid Kamal Abdelbasset , Dear editors,

thank you for your further interest in our manuscript entitled „Cortisol response under low intensity exercise during cognitive-behavioral therapy is associated with therapeutic outcome in panic disorder – an exploratory study“. We have now changed the title according to the reviewer´s suggestion and adapted the term „therapy outcome“ throughout the manuscript. All changes are marked yellow.

Please find attached our rebuttal letter with point-by-point replies (also marked yellow).

Do not hesitate to contact me in case of further inquiries.

Sincerely Yours,

Gloria Wintermann (on behalf of all co-authors)

Response: We have included a statement in the front page of our manuscript and also in the statistical analyses, concerning the availability of our data (see title page, see page 11, lines 258-260).

The data file and study protocol are available from the Open Science Framework (https://archive.org/details/osf-registrations-s76rw-v1, source https://osf.io/s76rw/, registration DOI 10.17605/OSF.IO/S76RW and identifier DOI 10.17605/OSF.IO/NE4G6).

Reviewer #2: The revised version of paper titled (Cortisol response under low intensity exercise during cognitive-behavioral therapy is

associated with therapy outcome in panic disorder – an exploratory study) is improved compared to the original submission.

I find it necessary that authors change (therapy outcome) int he title to (therapeutic outcome)

Thank you for this hint. We have adhered to your suggestion and replaced the word „therapy“ by „therapeutic“, throughout the whole manuscript, where appropriate.

---

## [Decision Letter · Decision Letter 2]

11 Jul 2022

PONE-D-22-01757R2Cortisol response under low intensity exercise during cognitive-behavioral therapy is associated with therapeutic outcome in panic disorder – an exploratory studyPLOS ONE

Dear Dr. Wintermann,

Thank you for submitting your manuscript to PLOS ONE. After careful consideration, we feel that it has merit but does not fully meet PLOS ONE’s publication criteria as it currently stands. Therefore, we invite you to submit a revised version of the manuscript that addresses the points raised during the review process.

We look forward to receiving your revised manuscript.

Kind regards,

Walid Kamal Abdelbasset, Ph.D.

Academic Editor

PLOS ONE

Journal Requirements:

Reviewers' comments:

Reviewer's Responses to Questions

**Comments to the Author**

1. If the authors have adequately addressed your comments raised in a previous round of review and you feel that this manuscript is now acceptable for publication, you may indicate that here to bypass the “Comments to the Author” section, enter your conflict of interest statement in the “Confidential to Editor” section, and submit your "Accept" recommendation.

Reviewer #1: (No Response)

Reviewer #2: All comments have been addressed

2. Is the manuscript technically sound, and do the data support the conclusions?

Reviewer #1: Partly

Reviewer #2: Yes

3. Has the statistical analysis been performed appropriately and rigorously? 

Reviewer #1: Yes

Reviewer #2: Yes

4. Have the authors made all data underlying the findings in their manuscript fully available?

Reviewer #1: Yes

Reviewer #2: Yes

5. Is the manuscript presented in an intelligible fashion and written in standard English?

Reviewer #1: Yes

Reviewer #2: Yes

6. Review Comments to the Author

Reviewer #1: The authors should include in the limitations section a statement to the effect that with the multiple statistical tests being performed in this limited sample that no adjustment to the p-value has been made for multiple testing and thus the p-values should be interpreted with caution. This was ignored in my previous review comment.

Reviewer #2: The revised form of the manuscript titled ( Cortisol response under low intensity exercise during cognitive-behavioral therapy is associated with therapeutic outcome in panic disorder – an exploratory study) was revised adequately according to the reviewers' recommendations.

7. PLOS authors have the option to publish the peer review history of their article (what does this mean?). If published, this will include your full peer review and any attached files.

Reviewer #1: No

Reviewer #2: **Yes: **Sawsan Zaitone

---

## [Author Response · Author response to Decision Letter 2]

11 Jul 2022

Dear ladies and gentlemen,

thank you for the opportunity to revise and profoundly improve our manuscript entitled „Cortisol response under low intensity exercise during cognitive-behavioural therapy is associated with therapeutic outcome in panic disorder – an exploratory study“ by Wintermann, G.-B., Noack, R., Schmiedgen, S. and Weidner, K.

We have tried to adhere to the reviewers´ requests and included the open points in the revised version of our manuscript. Changes are coloured.

Please feel free to contact us in case of any further inquiries.

Sincerely Yours,

Gloria Wintermann (on behalf of all co-authors)

2. Is the manuscript technically sound, and do the data support the conclusions?

Reviewer #1: Partly

We have addressed this point in the limitations of the study and made a suggestion for designing future studies in this research field (see lines 485-503).

Reviewer #2: Yes

Reviewer #1: The authors should include in the limitations section a statement to the effect that with the multiple statistical tests being performed in this limited sample that no adjustment to the p-value has been made for multiple testing and thus the p-values should be interpreted with caution. This was ignored in my previous review comment.

We have already addressed this point partly in the revised version of our manuscript. Bonferroni-correction was applied when bivariate correlations between psychopathology and cortsiol response, before and after the therapeutic intervention, were analysed (see lines 256-257). We have now added a comment, that results should be interpreted with caution and as preliminary since only a small sample size was enrolled (see lines 490-491). We have also made a suggestion for designing future sample sizes (lines 493-494). Above, we added a sentence that p-value adjustment for multiple testing was not applied in all statistical analyses (lines 495-496).

---

## [Editor Report · Decision Letter 3]

9 Aug 2022

Cortisol response under low intensity exercise during cognitive-behavioral therapy is associated with therapeutic outcome in panic disorder – an exploratory study

PONE-D-22-01757R3

Dear Dr. Wintermann,

We’re pleased to inform you that your manuscript has been judged scientifically suitable for publication and will be formally accepted for publication once it meets all outstanding technical requirements.

Kind regards,

Walid Kamal Abdelbasset, Ph.D.

Academic Editor

PLOS ONE

Additional Editor Comments (optional):

All comments have been addressed. No further comments are required.
---

## [Editor Report · Acceptance letter]

23 Aug 2022

PONE-D-22-01757R3 

Cortisol response under low intensity exercise during cognitive-behavioral therapy is associated with therapeutic outcome in panic disorder – an exploratory study 

Dear Dr. Wintermann:

I'm pleased to inform you that your manuscript has been deemed suitable for publication in PLOS ONE. Congratulations! Your manuscript is now with our production department. 

Kind regards, 

on behalf of

Dr. Walid Kamal Abdelbasset 

Academic Editor

PLOS ONE